# A Comprehensive Analysis of Clinical Trials in the COVID-19 Pandemic Era

**DOI:** 10.3390/medicina56060315

**Published:** 2020-06-26

**Authors:** Jinhee Lee, Han Wul Shin, Jun Young Lee, Jae Seok Kim, Jae Won Yang, Keum Hwa Lee, Andreas Kronbichler, Jae Il Shin

**Affiliations:** 1Department of Psychiatry, Yonsei University, Wonju College of Medicine, Wonju Kangwon 26426, Korea; jinh.lee95@yonsei.ac.kr; 2Department of Nephrology, Yonsei University, Wonju College of Medicine, Wonju Kangwon 26426, Korea; dragonshw@yonsei.ac.kr (H.W.S.); junyoung07@yonsei.ac.kr (J.Y.L.); ripplesong@yonsei.ac.kr (J.S.K.); kidney74@yonsei.ac.kr (J.W.Y.); 3Department of Pediatrics, Yonsei University College of Medicine, Yonsei-ro 50, Seodaemun-gu, C.P.O. Box 8044, Seoul 03722, Korea; AZSAGM@yuhs.ac; 4Department of Internal Medicine IV (Nephrology and Hypertension), Medical University Innsbruck, 6020 Innsbruck, Austria; andreas.kronbichler@i-med.ac.at

**Keywords:** COVID-19, clinical trials

## Abstract

*Background and objective:* Despite medical advances, we are facing the unprecedented disaster of the coronavirus disease 2019 (COVID-19) pandemic without available treatments and effective vaccines. As the COVID-19 pandemic has approached its culmination, desperate efforts have been made to seek proper treatments and response strategies, and the number of clinical trials has been rapidly increasing. In this time of the pandemic, it is believed that learning lessons from it would be meaningful in preparing for future pandemics. Thus, this study aims at providing a comprehensive landscape of COVID-19 related clinical trials based on the ClinicalTrials.gov database. *Materials and methods:* Up to 30 March 2020, we identified a total of 147 eligible clinical trials and reviewed the overview of the studies. *Results:* Until then, the most clinical trials were set up in China. Treatment approaches are the most frequent purpose of the registered studies. Chloroquine, interferon, and antiviral agents such as remdesivir, lopinavir, and ritonavir are agents under investigation in these trials. *Conclusions:* In this study, we introduced the promising therapeutic options that many researchers and clinicians are interested in, and to address the hidden issues behind clinical trials in this COVID-19 pandemic.

## 1. Introduction

After the first cluster of cases with coronavirus disease-19 (COVID-19) was reported in December 2019, the disease has been rapidly spreading around the world in an unprecedented way [1]. The number of infected patients has rapidly increased, eventually became pandemic that the World Health Organization declared as a public health emergency of international concern [2]. Unfortunately, there are still no efficient antiviral agents and vaccines for severe acute respiratory syndrome coronavirus 2 (SARS-CoV-2) infections. Thus, a large number of clinical trials have been initiated with massive efforts to identify effective therapeutic agents against COVID-19. The continually pouring over clinical trials represent what numerous researchers and clinicians are thinking and planning to fight against the pandemics including, COVID-19. Thus, it is believed that identifying and analyzing trends, and backgrounds of these clinical trials could provide valuable lessons to cope with other potential pandemics in the future.

For transparency in the entire research process, the International Committee of Medical Journal Editors (ICMJE) announced that clinical trials should be registered on the public site before conducting the research. After 1 July 2005, this policy has been applied to all clinical trials starting recruitment [3]. The National Library of Medicine (NLM) developed a registry (ClinicalTrials.gov) that provides the most effective source of information for all kinds of clinical research. However, most of the clinical trials require a lot of time, funding, and effort to complete them. Because of the rapid rate of virus transmission, an efficient strategy is needed to maximize public health benefits.

As ClinicalTrials.gov represents the latest research on COVID-19 to date, we aimed to analyze all of the COVID-19 related studies that have been conducted so far and to provide a comprehensive landscape of the COVID-19 pandemic.

## 2. Materials and Methods

### 2.1. Data Source and Eligible Study

Two physicians (J.Y.L. and H.W.S.) at the Yonsei University Wonju College of Medicine used the term “COVID 19” and “Sars-Cov2” to search all the registered clinical trials in the ClinicalTrials.gov database. All identified trials were recorded on Google drive (Google Corporation, Mountain View, CA, USA) as spreadsheet form, which makes it possible that all the recorded databases could be discussed at any time. The other physician (J.L.) would review the data recorded by the two physicians, and any disagreements were solved by consensus or referring to the other physician (J.I.S.) who has more than twenty years of experience in clinical trials. Up to 30 March 2020, a total of 172 trials were identified. After carefully reviewing all the information presented by the ClinicalTrials.gov database, 25 trials were excluded. One-hundred forty-seven trials were left for further analysis.

### 2.2. Study Variables

Before the performance of our search, we set up a record standards for each study variable and the following characteristics provided by ClinicalTrials.gov database were assessed: registration time, country of registration, type of study (interventional or observational, randomized controlled trial (RCT)), the purpose of the study (the outcome of the study), registration number, time perspective (prospective or retrospective), included participants (all COVID-19 confirmed patients, suspected person who did not have confirmed COVID-19, close contacted with COVID-19 confirmed patients who are tested negative for COVID-19, medical staff who have close contact with COVID-19 patients), type of treatment, primary and secondary outcome, masking (none or open-label or blind), allocation (none or randomized or non-randomized), study arm (none or one or two or more), funding source (university or hospital or industry or other), participant age, center (one or two or more), and participant country (single or two or more).

### 2.3. Statistical Analysis

The characteristics of clinical trials were summarized by descriptive statistics: continuous variables were characterized as mean and standard deviations (SD), and categorical variables were reported as numbers and percentages. When performing univariate analyses, Chi-square (for categorical variables) and independent *t*-test (for continuous variables) were used to compare the difference for the characteristics between the two groups. Fisher’s exact test was also applied if number of cases was small for categorical variables. All statistical tests were performed using with IBM Statistics Package for the Social Science (SPSS) version 25.0 (IBM Corporation, Armonk, NY, USA) and a two-sided *p* < 0.05 was considered statistically significant.

## 3. Results

### 3.1. Baseline Characteristics of Registered Clinical Trials

Among 172 trials from ClinicalTrials.gov, 25 trials were excluded because of duplication, not related to COVID-19, 9 about Chinese herbs, 16 disappeared when a second search was performed at ClinicalTrials.gov. Finally, 147 trials (63 RCTs, 29 Non-RCTs, 55 observational studies) were analyzed. As shown in Figure 1, there was an increasing trend in the number of registered studies over time. In the investigational period, most of the studies were registered in China, 34.7% (51) (Figure 2). 63 (42.9%) studies were classified as RCT, 29 (19.7%) studies were classified as non-RCT, and 55 (37.4%) studies were classified as observational studies. Three trials were registered as multi-country study, and 39 trials were registered as a multicenter study. Most of the studies (131, 88.5%) were studies conducted by hospitals or universities or non-governmental organizations (NGO). Analysis of the effectiveness of the treatment is most frequent purpose of the study (Figure 3).

Among various medications, chloroquine and hydroxychloroquine were the most frequently drugs under investigation (n = 16) (Figure 4). When classified according to therapeutic mechanisms, antiviral agents were the most studied treatment medication (n = 17) (Figure 5).

### 3.2. Studies Registered as Randomized Controlled Trials

All studies performed randomization, but only 35 (55.6%) studies performed blinding. A brief summary of enrolled studies’ characteristics is shown in Appendix A. Treatment effectiveness was the most frequent purpose of the enrolled studies (54, 85.7%), and prevention of COVID-19 was the next most frequent purpose of the enrolled studies (8, 12.7%). Among 38 kinds of various treatment medications (49 kinds of comparisons), hydroxychloroquine (6, 9.5%) was the most frequently studied medication. The clinical course was the most frequent primary and secondary outcomes of studies (33, 52.4%/31, 49.2%), the average number of participants was 1339.43 (minimum 20, maximum 40,000) and 11 trials were sponsored by industrial companies.

When performed univariate analyses, multicenter studies were associated with open labeled masking compared to blinding (*p* = 0.047). Single center studies used more blinding and included fewer multi-country studies than multicenter trials (*p* = 0.047 and *p* = 0.045, respectively). Multi-country studies had more likely sponsors from industries (*p* = 0.004, respectively). There was no difference between randomized controlled trials performed in China and outside China (Table 1).

### 3.3. Study Registered as Non-Randomized Controlled Trials

All studies were prospective nonblinded. Twenty (69%) studies investigated treatment effectiveness (Appendix A). Fourteen (48.3%) trials were registered in China and eight trials (27.6%) were registered in Italy. Clinical assessment of the disease course is the most frequent primary outcome of the studies (10, 34.5%) and the same was the case for the secondary outcome (9, 31%). There were 24 kinds of treatments in non-randomized trials and mesenchymal stem cell treatment is the leading intervention. Seven trials were multicenter studies and four trials were sponsored by the industry. There were no multi-country studies. The mean of participant’s number of patients was 253.72 (minimum 4, maximum 2944).

In univariate analyses, single group (n = 18, 90.0%) showed more single arm than parallel group (n = 1, 11.1%) (*p* < 0.001). Other analyses showed no difference between groups such as China/outside of China, multicenter and sponsored group (Table 2).

### 3.4. Study Registered as Observational Trials

Among 55 observational studies, 11 were retrospective (20%), while the others were prospective. Most of the trials were registered in China and Italy (19, 35%/13, 23.6%). Treatment was the most frequent purpose of the respective study (23, 41.8%) and the diagnosis was the next frequent purpose of the study (21, 38.2%). The clinical course was the most frequent primary and secondary outcomes of the trials (18, 32.7%/17, 30.9%). Eight trials were multicenter and two were industry sponsored (Appendix A).

In Table 3, there was only a significance between countries (China or outside of China) and direction (prospective or retrospective). In other words, countries outside of China (n = 32, 91.4%) performed more observational trials than China (n = 12, 60.0%) (*p* = 0.011). Multicenter and group of trials with a sponsor did not have statistically difference.

## 4. Discussion

In the COVID-19 pandemic era, numerous clinical trials are in place and are scheduled to be implemented. In particular, since there are no effective antiviral drugs and vaccines for COVID-19, many drugs and interventions are being tested in various ways [4] and while the number of COVID-19 is peaking in most countries, the greater the number of these clinical trials is. Currently, most clinical trials registered at Clinicaltrials.gov are interventional studies to identify the effects of existing drugs on COVID-19 and many follow-up RCTs are planned and performed to obtain clear and reliable evidence of clinical effectiveness.

Treatment approaches for COVID-19 now show an extensive range of diversity, ranging from traditional Chinese herbs to stem cell therapy. The diversity and rapid increase of these clinical trials may be primarily due to the urgency of not having a realistic treatment for COVID-19. However, in another aspect, the urgency of the pandemic has resulted in simplified clinical research as the barriers to perform clinical trials have been lowered and sometimes, if unintentionally, strict ethical demands have been eased. In regard to the trends for the treatment of COVID-19, our study shows that chloroquine and hydroxychloroquine were the most frequently studied agents (Figure 4) [5,6,7,8,9]. Chloroquine and hydroxychloroquine have been used in the treatment of malaria, providing extensive clinical experiences. In addition, it can be administered orally and is inexpensive. In particular, they can be used to not only treat but also prevent COVID-19 [10], which may be the reason many researchers choose them first as a targeted drug. However, many studies do not investigate the efficacy of the drug [11]. Recently, there has been a report of side effects, including the occurrence of severe arrhythmia, associated with the use of hydroxychloroquine in the treatment of patients with COVID-19 [12]. Such report seems to pour cold water on the promising plans of the early development of the cheap and effective drugs for COVID-19, but at the same time, the study also aroused the necessity of care and prudence in the overheated competition of clinical trials.

The next most widely studied therapeutic agents are interferon (Figure 4). Interferon has a potential of directly inhibiting viral replication and the roles of interferon have been already proven in many clinical diseases, especially in viral hepatitis. Thus, the therapeutic role of interferon was studied even in severe acute respiratory syndrome (SARS) and middle east respiratory syndrome (MERS) pandemics. A few positive results were reported in in vitro and in vivo studies [13], while a following meta-analysis study did not provide concrete evidence [14], its clinical effects remaining inconclusive up to date. Now, a few clinical trials are ongoing in which interferon are being applied either alone or as a combination therapy with antiviral agent.

Antiviral agents are the most studied treatment option in this COVID-19 pandemic, although there are fewer studies compared to hydroxychloroquine and interferon as individual drug (Figure 5). Various antiviral drugs have already been examined for their efficacy as a therapeutic agent in SARS and MERS [15]. Although no antiviral drugs have a conclusive basis as a therapeutic agent, it seems likely that drugs that have accumulated various data from existing studies are considered primarily as a candidate for a new therapeutic agent in the COVID-19 pandemic. Moreover, the demand for antiviral drugs that directly inhibit the growth of viruses with more relevant mechanisms is still high [16]. The most commonly studied antiviral drug for COVID-19 is lopinavir and ritonavir. Both are used to treat acquired immune deficiency syndrome (AIDS) have been mainly studied to verify the effects of combination therapy rather than a single treatment. This trend seems to reflect the general perception of clinicians that these drugs are limited in effectiveness, either alone or even as a combination of two drugs [17]. Remdesivir is also an interesting antiviral agent that is currently considered to be the most promising treatment for COVID-19. Remdesivir has shown effects in non-human studies of SARS and MERS [18,19], while there is not enough clinical data in human studies yet. The recent results of an interventional study investigating clinical effects of remdesivir showed the potential for the treatment of COVID-19 [20]. In addition, interim results of study conducted by National Institutes of Health’s National Institute of Allergy and Infectious Disease (NIAID) showed that remdesivir had 31% faster time to recovery and survival benefit. We are waiting for further results, while US Food and Drug Administration (USFDA) has already approved the emergency use of remdesivir in consideration of the urgency of the COVID-19 pandemic. A reliable study is needed in the end and further RCTs are currently underway to clearly determine the effectiveness of remdesivir.

Another feature of the clinical trials being implemented in the management of COVID-19 is that several novel therapeutic agents are targeted therapies [21]. In particular, the clinical trials of cytokine blockade such as tocilizumab, sarilumab, anakinra, and emapalumab are being tried to mitigate the cytokine storm in COVID-19 [22,23,24,25]. On the other hand, a few traditional anti-inflammatory drugs are still attracting attention. In particular, steroids have been used empirically to treat serious conditions even in SARS and MERS [26,27,28,29]. Despite the controversy of the efficacy and side effects [30,31], steroids have been widely used in actual clinical practice and clinical studies are also being conducted to verify the effects of steroids in the COVID-19 pandemic.

It is somewhat unexpected that studies on the use of colchicine in the treatment of COVID-19 are not small. Colchicine is non-selective inhibitor of NLRP3 inflammasomes and mitigating interleukin activation [32]. Colchicine has been mainly used as an anti-inflammatory drug in gouty arthritis and this anti-inflammatory action is also likely to have a useful effect in the treatment of COVID-19 [33]. In the end, many researchers seem to believe that adequate control of excessive systemic inflammation, such as cytokine storm is an important therapeutic goal in the treatment of COVID-19 as much as a direct treatment against coronavirus [34].

Interestingly, there are a few clinical trials of stem cell therapy for the treatment of COVID-19 [35]. In fact, stem cell therapy does not yet have solid clinical utility and clear indications, while the above-mentioned drugs have shown successful clinical outcomes in some clinical fields. Nevertheless, the phenomena of active clinical trials of stem cell therapy, along with the preference of novel targeted therapies, illustrate the adventurous trend among researchers and clinicians for seeking novel therapeutic agents in the pandemics without the treatment.

In the current circumstances, most clinical trials of COVID-19 appear to be led by hospitals, universities, or public research institutes. In fact, many biotech and pharmaceutical companies are focusing on the clinical application of products such as drugs or diagnostic kits produced by their own companies. However, at a time when faced with public disasters such as this COVID-19 pandemic, the excellent material and human resources of each commercial company must be a huge asset if well organized and properly used. Therefore, it is still regrettable that the role led by commercial companies in many clinical trials is relatively small, even though it can be understood that there is a fundamental difference between the aim of trials and the role of these companies. It is expected to set up a highly efficient response system to the pandemic if the coordinator is able to compromise the needs for excellent material and human resources and the commercial interests of the companies. Recently, several car manufacturers have joined to solve the problem of the shortage of mechanical ventilators in the United States. Working together to play a public role to the extent that it does not infringe upon the private autonomy of the companies is thought to be a good model for preparing another infectious pandemic in the future.

RCT studies in the COVID-19 pandemic still seem to be accompanied by ethical issues as was the case during other pandemics. For a long time, there have been discussions about the ‘principle of clinical equipoise’ in RCT [36,37,38]. Most RCTs are being tried to find a more reliable basis when some drugs or interventions are recognized to be effective to some extent through previous interventional studies. Therefore, the drugs or interventions that are planned for RCT may already be expected to work, although they have no concrete evidence of efficacy yet. In this regard, RCTs that start with such a premise can hardly be seen as truly adhering to the ‘principle of clinical equipoise’. Nevertheless, the problem is, in fact, there is no proper alternative. In the face of the real problem of the pandemic, we are racing to find the key for a cure. In the process, those problems leave us with an ethical dilemma. Some studies set up a control group on which basic treatments are essentially applied in design. In addition, they set up a treatment group by adding targeted drugs or interventions. This type of research may alleviate some of the ethical dilemmas.

## 5. Conclusions

In this study, we outlined but extensively examined the rapidly changing clinical trials in the COVID-19 pandemic. It is important to know which drugs are of high interest and which treatments are newly introduced. However, at the same time, through looking at the trends of various clinical trials, understanding why certain treatments are preferentially chosen by many researchers and discussing the hidden issues behind clinical trials in the pandemic era is also crucial to prepare for future pandemics. Above all, we believe that it would be good if we could establish and implement practical response strategies that can complement emergent elements with ethical ones without conflict in this COVID-19 pandemic.

Our study has an important limitation. Despite of our efforts, many studies were added since we started this study, and our study still does not have much recent research information. Not all data registered for clinicaltrials.gov are continued; some trials did not recruit patients over several months or disappeared when re-checked on the site. Research in other areas besides therapeutic agents is difficult to identify in detail, due to the nature of characteristics of site; therefore, the analysis through the site is limited. Nevertheless, the overall trends of trials are thought to be unchanged and we believe that our discussions on essential issues are still valid.

## Figures and Tables

**Figure 1 medicina-56-00315-f001:**
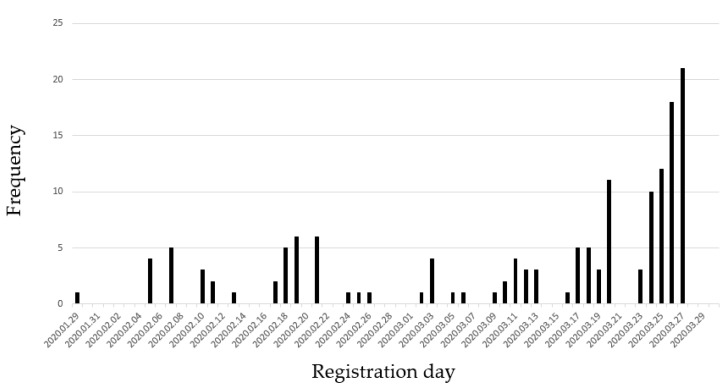
Number of enrolled studies according to date (from 29 January 2020 to 30 March 2020).

**Figure 2 medicina-56-00315-f002:**
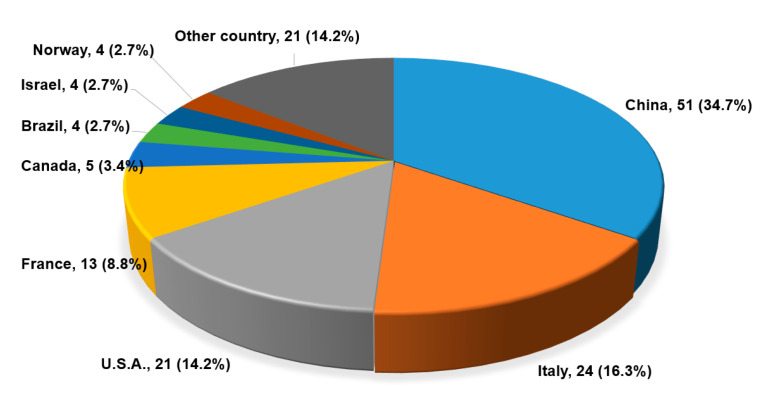
Distribution of countries registered at cinicaltrials.gov.

**Figure 3 medicina-56-00315-f003:**
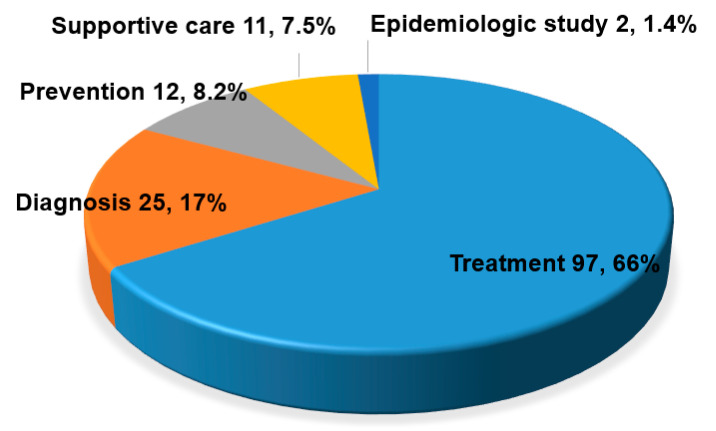
Purpose of study registered at clinicaltrials.gov.

**Figure 4 medicina-56-00315-f004:**
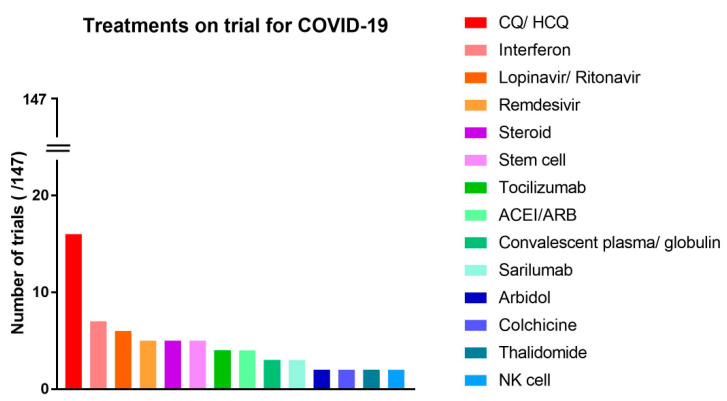
Treatments on trials for COVID-19, listed with individual drugs. (ACEI, angiotensin-converting enzyme inhibitor; ARB, angiotensin receptor blocker; CQ, chloroquine; HCQ, hydroxychloroquine; NK, natural killer).

**Figure 5 medicina-56-00315-f005:**
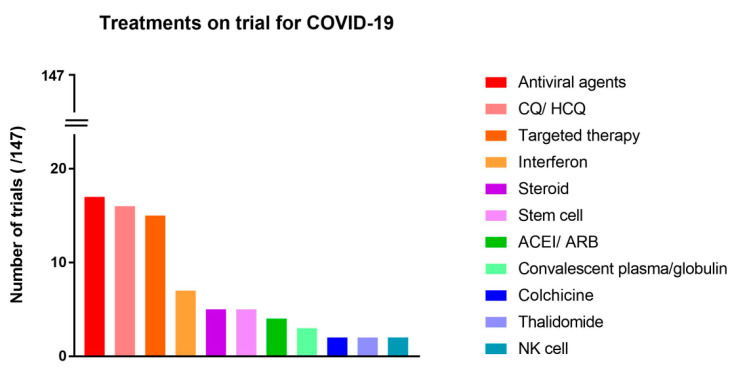
Treatments on trial for COVID-19, listed with therapeutic mechanisms. (ACEI, angiotensin-converting enzyme inhibitor; ARB, angiotensin receptor blocker; CQ, Chloroquine; HCQ, hydroxychloroquine; NK, natural killer).

**Table 1 medicina-56-00315-t001:** Clinical characteristics in randomized controlled studies (RCT).

	**China (n = 19)**	**Studies Outside China (n = 44)**	***p*-Value**
Planned Enrollment	136.2 ± 114.6	1944.9 ± 6109.6	0.204
Blinded	11 (57.9%)	24 (54.5%)	0.806
Multicenter	7 (36·8%)	16 (36·4%)	0.971
Multi-Country	0 (0%)	3 (6·8%)	0.547
Sponsor (Industry)	1 (5.3%)	10 (22.7%)	0.150
	**Blinded (n = 35)**	**Open Label (n = 28)**	
Planned Enrollment	1933.8 ± 6817.7	731.4 ± 1303.5	0.362
China	11 (31.4%)	8 (28.6%)	0.806
Multicenter	9 (25.7%)	14 (50.0%)	0.047
Multi-Country	1 (2.9%)	2 (7.1%)	0.560
Sponsor (Industry)	7 (20.0%)	4 (14.3%)	0.553
	**Multicenter (n = 23)**	**Single Center (n = 40)**	
Planned enrollment	959.9 ± 1719.9	1652.2 ± 6358.2	0.612
China	7 (30.4%)	12 (30.0%)	0.971
Blinded	9 (39.1%)	26 (65.0%)	0.047
Multi-Country	3 (13.0%)	0 (0%)	0.045
Sponsor (Industry)	5 (21.7%)	6 (15.0%)	0.498
	**Multi-Country (n = 3)**	**Single Country (n = 60)**	
Planned Enrollment	2573.3 ± 3058.5	1340.7 ±5249.2	0.690
China	0 (0%)	19 (31.7%)	0.547
Blinded	1 (33.3%)	34 (56.7%)	0.427
Multicenter	3 (100%)	20 (33.3%)	0.045
Sponsor (Industry)	3 (100%)	8 (13.3%)	0.004
	**Sponsor (Industry) (n = 11)**	**Researcher (n = 52)**	
Planned Enrollment	905.8 ± 1744.6	1503.9 ± 5627.4	0.730
China	1 (9.1%)	18 (34.6%)	0.150
Blinded	7 (63.6%)	28 (53.8%)	0.741
Multicenter	5 (45.5%)	18 (34.6%)	0.511
Multi-Country	3 (27.3%)	0 (0%)	0.004

Data are mean ± SD or number (percentage) of patients.

**Table 2 medicina-56-00315-t002:** Clinical characteristics in non-randomized controlled studies (non-RCT).

	**China (n = 14)**	**Outside of China (n = 15)**	***p*-Value**
Participant	260.0 ± 775.0	247.9 ± 294.9	0.955
Multicenter	3 (21.4%)	4 (26.7%)	1.000
Sponsor (Industry)	2 (14.3%)	2 (13.3%)	1.000
Single Group	10 (71.4%)	3 (66·7%)	1.000
Single Arm	10 (74.4%)	9 (60.0%)	0.700
	**Multicenter (n = 7)**	**Single Center (n = 22)**	
Participant	354.3 ± 336.0	221.7 ± 627.0	0.600
China	3 (42.9%)	11 (50.0%)	1.000
Sponsor (Industry)	1 (14.3%)	3 (13.6%)	1.000
Single Group	5 (71.4%)	15 (68.2%)	1.000
Single Arm	5 (71.4%)	14 (63.6%)	1.000
	**Sponsor (Industry) (n = 4)**	**Researcher (n = 25)**	
Participant	299.5 ± 468.3	246.4 ± 590.2	0.866
China	2 (50.0%)	12 (48.0%)	1.000
Multicenter	1 (25.0%)	6 (24.0%)	1.000
Single Group	2 (50.0%)	18 (72.0%)	0.568
Single Arm	1 (25.0%)	18 (72.0%)	0.105
	**Single Group (n = 20)**	**Parallel Group (n = 9)**	
Participant	122.9 ± 170.4	544.6 ± 959.9	0.226
China	10 (50.0%)	4 (44.4%)	1.000
Multicenter	5 (25.0%)	2 (22.2%)	1.000
Sponsor (Industry)	2 (10.0%)	2 (22.2%)	0.568
Single Arm	18 (90.0%)	1 (11.1%)	<0.001
	**Single Arm (Industry) (n = 19)**	**Non-Randomized (n = 10)**	
Participant	151.1 ± 203.5	448.7 ± 925.1	0.340
China	10 (52.6%)	4 (40.0%)	0.700
Multicenter	5 (26.3%)	2 (20.0%)	1.000
Sponsor (Industry)	1 (5.3%)	3 (30.0%)	0.105
Single Group	18 (94.7%)	2 (20.0%)	<0.001

Data are mean ± SD or number (percentage) of patients.

**Table 3 medicina-56-00315-t003:** Clinical characteristics in observational study.

	**China (n = 20)**	**Outside of China (n = 35)**	***p*-Value**
Participant	17,264.1 ± 66,918.8	6494.9 ± 20,004.3	0.377
Prospective	12 (60.0%)	32 (91.4%)	0.011
Multicenter	4 (20.0%)	4 (11.4%)	0.443
Sponsor (Industry)	1 (5.0%)	1 (2.9%)	1.000
	**Prospective (n = 44)**	**Retrospective (n = 11)**	
Participant	12,849.8 ± 47,942.9	655.6 ± 1454.2	0.406
China	12 (27.3%)	8 (72.7%)	0.011
Multicenter	7 (15.9%)	1 (9.1%)	1.000
Sponsor (Industry)	2 (4.5%)	0 (0%)	1.000
	**Multicenter (n = 8)**	**Single Center (n = 47)**	
Participant	10,089.8 ± 19,219.4	10,465.7 ± 46,057.5	0.982
China	4 (50.0%)	16 (34.0%)	0.443
Prospective	7 (87.5%)	37 (78.7%)	1.000
Sponsor (Industry)	0 (0%)	2 (4.3%)	1.000
	**Sponsor (Industry) (n = 2)**	**Researcher (n = 53)**	
Participant	210.0 ± 127.3	10,795.9 ± 43,842.0	0.736
China	1 (50.0%)	19 (35.8%)	1.000
Prospective	2 (100%)	42 (79.2%)	1.000
Multicenter	0 (0%)	8 (15.1%)	1.000

Data are mean ± SD or number (percentage) of patients.

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
