# Peer review of "A Comprehensive Analysis of Clinical Trials in the COVID-19 Pandemic Era"

_medicina, 2020, doi:10.3390/medicina56060315_

Round 1

Reviewer 1 Report

The authors present a database analysis based on a clinicaltrials.gov search.

Their primary aim is to describe the planned and active clinical trials related to the current COVID19 pandemic.

Overall, this manuscript reaches its aim to provide a descriptive overview on registered trials.

The following comments should be addressed by the authors:

Section 2.1.:  The database search was done by searching for the term "COVID 19". clinicaltrials.gov automatically might have exteded this by related terms e.g. "Sars-Cov2". Please check and also add this information into your manunscript.

Section 2.3.: The sentence "any missing data were excluded" is unclear. Did you mean that studies with missing data were not included into the analysis?

Section 3.1: Within the first sentence the term "article" is used for "trials" or "studies". Please change accordingly.

Can you please explain how 16 trials disappeared from your second search?
Because retracted trials will likely be findable but usually does not disappear, I would guess this has something to do with the search strategy.

Figure 1: Please change the x-axis to a better representation of the time scales. Currently, distances (in terms of time intervals) between the bars can be different. Please consider a line graph over an actual time-axis. Alternatively, a cumulative line graph with cumulative number of trials might be informative.

Figure 2: The title can be deleted from the actual figure. It should be sufficient to have it in the figure caption. Also, the title /caption is misleading. This is not the number of countries registered at clinicaltrials.gov. It's the distribution of countries with registered COVID19 trials.

Figure 3: Same - please delete the title from the figure.

Section 3.2: You separated your trials by RCT  or non RCT or observational and analysed these separatedly.

Please reconsider the term "total number". You mean "Planned enrollment" probably? This might be the better term, since trials are not finished yet and we do not know the final sample size.

The discussion and conclusion is a very well written summary and sets the mainly descriptive work into a good context.

Author Response

Dear Editor

First of all, we sincerely appreciate for all the comments that you gave us to improve the manuscript. We all agree with your comments entirely, and have tried to correct or improve the manuscript according to your suggestions. Here, we presents the details of the correction as follows.

Reviewer 1

Section 2.1.:  The database search was done by searching for the term "COVID 19". clinicaltrials.gov automatically might have exteded this by related terms e.g. "Sars-Cov2". Please check and also add this information into your manunscript.

We researched clinicaltrials.gov not only “COVID 19” but also “Sars-Cov2“

We add this information into revised manuscript with track change “Two physicians (JYL and HWS) at the Yonsei University Wonju College of medicine used the term “COVID 19” and “ Sars-Cov2”to search all the registered clinical trials in the ClinicalTrials.gov database ”

Section 2.3.: The sentence "any missing data were excluded" is unclear. Did you mean that studies with missing data were not included into the analysis?

On statistical analysis, there was no missing date. Therefore, we erased “any missing data were excluded” at revised manuscript with track change.

Section 3.1: Within the first sentence the term "article" is used for "trials" or "studies". Please change accordingly.

 According to reviewer’s comments we changed article to trials in the manuscript with track change

Can you please explain how 16 trials disappeared from your second search? 
Because retracted trials will likely be findable but usually does not disappear, I would guess this has something to do with the search strategy.

To register data, we searched trials on clinicaltrials.gov every day until final analysis. However, on double checking at the final analysis, some trials were not found at the clinicaltrial.gov. Therefore, we researched all trials to check the current status of trials. Finally, 16 trials disappeared at the final analysis.

Figure 1: Please change the x-axis to a better representation of the time scales. Currently, distances (in terms of time intervals) between the bars can be different. Please consider a line graph over an actual time-axis. Alternatively, a cumulative line graph with cumulative number of trials might be informative.

 According to reviewer’s comment we changed figure 1

Figure 2: The title can be deleted from the actual figure. It should be sufficient to have it in the figure caption. Also, the title /caption is misleading. This is not the number of countries registered at clinicaltrials.gov. It's the distribution of countries with registered COVID19 trials.

According to reviewer’s comment we deleted title from the actual figure and changed in the manuscript with track change

Figure 3: Same - please delete the title from the figure.

 According to reviewer’s comment we deleted title from the actual figure and changed in the manuscript with track change

Section 3.2: You separated your trials by RCT  or non RCT or observational and analysed these separatedly.

Please reconsider the term "total number". You mean "Planned enrollment" probably? This might be the better term, since trials are not finished yet and we do not know the final sample size.

According to reviewer’s comment we changed from total number to planned enrollment at the revised manuscript with track change

The discussion and conclusion is a very well written summary and sets the mainly descriptive work into a good context.

We would be very pleased if our review is suitable for publication in your journal.

Thank you for your great helps! Have a wonderful day!

Best regards,

JAE IL SHIN

Reviewer 2 Report

I preliminarily declare a potential CoI as a co-author of a similar article published in Allergy :

COVID-19 clinical trials: quality matters more than quantity Bonini S, Maltese G Allergy 2020: May 20 https://doi.org/10.1111/all.14409 ; https://onlinelibrary.wiley.com/doi/abs/10.1111/all.14409

The article is well-written and methods are appropriate. Conclusions are sound.

The limitation of the paper - also recognized by authors - is that the review of studies listed by clinicaltrials.gov was made on March 31, 2020. The number of registered trials in the US registry, as well as in other registries, changes every day, and the actual figures are quite different from those reported in the article. Furthermore, in the meantime, several other similar articles were published, apart from our :

Randomized Clinical Trials and COVID-19
Managing Expectations
Howard Bauchner, MD; Phil B. Fontanarosa, MD, MBA; JAMA Published online May 4, 2020 doi:10.1001/jama.2020.8115
Coronavirus drugs trials need scale and collaboration. Editorial Nature | Vol 581 | 14 May 2020; p 120
Pharmacologic Treatments for Coronavirus Disease 2019 (COVID-19): A Review. Sanders JM, Monogue ML, Jodlowski TZ, Cutrell JB.JAMA. 2020 Apr 13. doi: 10.1001/jama.2020.6019. Online ahead of print.

This limitation - for which authors are not responsible for - reduces the novelty of the paper. This type of article would deserve a fast-track review and, if accepted, immediate publication. 

Apart from the above, there are no major or minor comments from my side.

Author Response

Dear Editor

First of all, we sincerely appreciate for all the comments that you gave us to improve the manuscript. We all agree with your comments entirely, and have tried to correct or improve the manuscript according to your suggestions. Here, we presents the details of the correction as follows.

I preliminarily declare a potential CoI as a co-author of a similar article published in Allergy :

COVID-19 clinical trials: quality matters more than quantity Bonini S, Maltese G Allergy 2020: May 20 https://doi.org/10.1111/all.14409 ; https://onlinelibrary.wiley.com/doi/abs/10.1111/all.14409

The article is well-written and methods are appropriate. Conclusions are sound.

The limitation of the paper - also recognized by authors - is that the review of studies listed by clinicaltrials.gov was made on March 31, 2020. The number of registered trials in the US registry, as well as in other registries, changes every day, and the actual figures are quite different from those reported in the article. Furthermore, in the meantime, several other similar articles were published, apart from our :

Randomized Clinical Trials and COVID-19
Managing Expectations
Howard Bauchner, MD; Phil B. Fontanarosa, MD, MBA; JAMA Published online May 4, 2020 doi:10.1001/jama.2020.8115

Coronavirus drugs trials need scale and collaboration. Editorial Nature | Vol 581 | 14 May 2020; p 120

Pharmacologic Treatments for Coronavirus Disease 2019 (COVID-19): A Review. Sanders JM, Monogue ML, Jodlowski TZ, Cutrell JB.JAMA. 2020 Apr 13. doi: 10.1001/jama.2020.6019. Online ahead of print.

This limitation - for which authors are not responsible for - reduces the novelty of the paper. This type of article would deserve a fast-track review and, if accepted, immediate publication. 

Apart from the above, there are no major or minor comments from my side

We totally agree about the reviewer’s comments, Therefore, we changed title as follow: “A comprehensive analysis of clinical trials in the COVID-19 pandemic era”

Reviewer 3 Report

The authors present an interesting study trying to summarizing currently active trials registered on clinicaltrials.gov. They claim to have searched clinicaltrials.gov up to March 30.

However, trials evaluating colchicine are not mentioned. To date more than 11 (eleven) studies have been announced on clinicaltrials.gov with three of them (COLCORONA NCT04322682; ColCOVID-19 - NCT04322565; GRECCO-19 - NCT04326790 – rationale and design published in Hellenic J Cardiol. 2020 Apr 3;S1109-9666(20)30061-0. doi: 10.1016/j.hjc.2020.03.002) having been announced up to the 30th of March.

Further, there is a current discussion on colchicine potential on COVID-19 patients is also discussed by a series of articles with focus on the hypothetical pathophysiological mechanisms have been published (e.g. Eur Heart J Cardiovasc Pharmacother. 2020 Apr 27:pvaa033. doi: 10.1093/ehjcvp/pvaa033 ; Clin Rheumatol. 2020 May 11:1-2. doi: 10.1007/s10067-020-05144-x. Online ahead of print; Hellenic J Cardiol. 2020 Apr 3;S1109-9666(20)30061-0. doi: 10.1016/j.hjc.2020.03.002)

Since proper and objective summary of the clinicaltrials registered studies is the main purpose of this article I suggest that  the aforementioned studies on colchicine should be included and discussed in the manuscript.

Author Response

Dear Editor

First of all, we sincerely appreciate for all the comments that you gave us to improve the manuscript. We all agree with your comments entirely, and have tried to correct or improve the manuscript according to your suggestions. Here, we presents the details of the correction as follows.

Since proper and objective summary of the clinicaltrials registered studies is the main purpose of this article I suggest that the aforementioned studies on colchicine should be included and discussed in the manuscript.

According to reviewer’s comment we added studies about colchicine at the discussion in the revised manuscript with track change

 It is somewhat unexpected that studies on the use of colchicine in the treatment of COVID-19 are not stmall. Colchicine has been mainly used as an anti-inflammatory drug in gouthy arthritis, and this anti-inflammatory action is also likely to have a useful effect in the treatment of COVID-19. In the end, many researchers seem to believe that adequate control of excessive systemic inflammation, such as cytokine storm, is an important therapeutic goal in the treatment of COVID-19 as much as a direct treatment against coronavirus

We would be very pleased if our review is suitable for publication in your journal.

Thank you for your great helps! Have a wonderful day!

Best regards,

JAE IL SHIN

Round 2

Reviewer 3 Report

Authors included "Colchicine" studies in their charts along with a paragraph in the discussion section. I would expect "Hellenic J Cardiol. 2020 Apr 3;S1109-9666(20)30061-0. doi: 10.1016/j.hjc.2020.03.002" to be cited as well along with a more detailed description of potential mechanisms of action (described in the aforementioned manuscript).

Author Response

Authors included "Colchicine" studies in their charts along with a paragraph in the discussion section. I would expect "Hellenic J Cardiol. 2020 Apr 3;S1109-9666(20)30061-0. doi: 10.1016/j.hjc.2020.03.002" to be cited as well along with a more detailed description of potential mechanisms of action (described in the aforementioned manuscript).

According to reviewer’s comment, we cited "Hellenic J Cardiol. 2020 Apr 3;S1109-9666(20)30061-0. doi: 10.1016/j.hjc.2020.03.002" and added potential mechanisms of action of colchicine in discussion section

It is somewhat unexpected that studies on the use of colchicine in the treatment of COVID-19 are not small. Colchicine is non-selective inhibitor of NLRP3 inflammasomes and mitigating interleukin activation [32]. Colchicine has been mainly used as an anti-inflammatory drug in gouthy arthritis, and this anti-inflammatory action is also likely to have a useful effect in the treatment of COVID-19 [33]. In the end, many researchers seem to believe that adequate control of excessive systemic inflammation, such as cytokine storm, is an important therapeutic goal in the treatment of COVID-19 as much as a direct treatment against coronavirus [34].

This manuscript is a resubmission of an earlier submission. The following is a list of the peer review reports and author responses from that submission.